# Additive Effect of Releasing Sterile Insects Plus Biocontrol Agents against Fruit Fly Pests (Diptera: Tephritidae) under Confined Conditions

**DOI:** 10.3390/insects14040337

**Published:** 2023-03-30

**Authors:** Pablo Montoya, Erick Flores-Sarmiento, Patricia López, Amanda Ayala, Jorge Cancino

**Affiliations:** 1Instituto de Biociencias, Universidad Autónoma de Chiapas, Boulevard Akishino S/N, Tapachula 30798, Chiapas, Mexico; 2Programa Moscas de la Fruta Senasica-Sader, Camino a Los Cacaotales S/N, Metapa de Domínguez 30860, Chiapas, Mexico; erick.flores.i@senasica.gob.mx (E.F.-S.); olga.lopez.i@senasica.gob.mx (P.L.); amanda.ayala.i@senasica.gob.mx (A.A.); jorge.cancino.i@senasica.gob.mx (J.C.)

**Keywords:** sterile insect technique (SIT), augmentative biological control (ABC), *Anastrepha ludens*, *Diachasmimorpha longicaudata*, *Coptera haywardi*, additive effect

## Abstract

**Simple Summary:**

Theoretically, the joint use of augmentative biological control (ABC) and sterile insects (SIT) can generate an additive or synergistic effect on the control of fruit fly pests. Field cage evaluations demonstrated that the sequential use of both techniques on a confined population of *Anastrepha ludens* lead to greater suppression than each technique acting alone. This was observed with the use of the larval parasitoid *Diachasmimorpha longicaudata* (parasitism ~70%), as well as the pupal parasitoid *Coptera haywardi* (~35% parasitism), although the high parasitism percentages by *D. longicaudata* contributed a greater suppressive effect on the fly population. The joint use of ABC and SIT induced ~70% sterility in the *A. ludens* population; however, the impact on the reproduction parameters of the fly population suggest that the joint action of both techniques was synergistic. Decreases in the fertility rate and hatching percentage of *A. ludens* eggs resulted in a lower intrinsic rate of increase (up to 50%) in the fly population.

**Abstract:**

Pest control models integrating the use of the sterile insect technique (SIT) and augmentative biological control (ABC) have postulated that it is possible to obtain a synergistic effect from the joint use of these technologies. This synergistic effect is attributed to the simultaneous attack on two different biological stages of the pest (immature and adult flies), which would produce higher suppression on the pest populations. Here we evaluated the effect of the joint application of sterile males of *A. ludens* of the genetic sexing strain Tap-7 along with two parasitoid species at the field cage level. The parasitoids *D. longicaudata* and *C. haywardi* were used separately to determine their effect on the suppression of the fly populations. Our results showed that egg hatching percentage was different between treatments, with the highest percentage in the control treatment and a gradual reduction in the treatments with only parasitoids or only sterile males. The greatest induction of sterility (i.e., the lowest egg hatching percentage) occurred with the joint use of ABC and SIT, demonstrating that the earlier parasitism caused by each parasitoid species was important reaching high levels of sterility. Gross fertility rate decreased up to 15 and 6 times when sterile flies were combined with *D. longicaudata* and *C. haywardi*, respectively. The higher parasitism by *D. longicaudata* was determinant in the decrease of this parameter and had a stronger effect when combined with the SIT. We conclude that the joint use of ABC and SIT on the *A. ludens* population had a direct additive effect, but a synergistic effect was observed in the parameters of population dynamics throughout the periodic releases of both types of insects. This effect can be of crucial importance in the suppression or eradication of fruit fly populations, with the added advantage of the low ecological impact that characterizes both techniques.

## 1. Introduction

Pest control models integrating the use of the sterile insect technique (SIT) and augmentative biological control (ABC) have postulated that it is possible to obtain a synergistic effect from the joint use of both technologies [1,2,3]. This synergistic effect may be due to the simultaneous attack on two different biological stages of the pest (immature and adult flies), which would lead to a more effective suppression of the pest populations [1,2,3]. The SIT has been used in different action programs against fruit fly pest populations around the world for eradication, containment, suppression, or prevention purposes [4]. A notable case was the eradication of the Mediterranean fly, *Ceratitis capitata* (Wied.) (Diptera: Tephritidae), on the border between Mexico and Guatemala [5,6,7,8]. There is also evidence of a successful use of the SIT for suppression purposes, such as in the case against the Mexican fruit fly, *Anastrepha ludens* (Loew), in the Soconusco region of the state of Chiapas in Mexico [9].

Several authors have shown that ABC through periodic releases of parasitoids is an efficient strategy for the suppression of fruit fly populations [10,11,12,13,14], showing that it is possible to reach parasitism percentages between 40 and 80%, depending on the host fruit dominating in the release zone. The use of these two control techniques against fruit fly pests offers great advantages in the current global scenario of climate change and the contamination of ecosystems, where two notable advantages are the null or little negative effect on the environment and the high specificity of these techniques [15]. The latter results in both techniques having a highly desirable profile for their use in the suppression or eradication of fruit flies under an area-wide approach.

The theoretical proposal of simultaneously using the SIT and the augmentative release of parasitoids has been experimentally validated by different authors. Wong et al. [16] demonstrated in 1992 in the Hawaiian Islands that concurrent releases of the parasitoid *Diachasmimorpha tryoni* (Cameron) (Hymenoptera: Braconidae) and sterile males to control *C. capitata* populations resulted in 47.2% parasitism and 75% reduction in egg hatching, which highly differed from the control area. Vargas et al. [17] observed that simultaneous releases of the parasitoid *Psyttalia fletcheri* (Silvestri) (Hymenoptera: Braconidae) and sterile males of the melon fly *Bactrocera cucurbitae* (Coquillett) in field cages resulted in suppression of 90% with respect to the control treatment. In Guatemala, the release of sterile males of *C. capitata* plus a pair of parasitoid species, the egg parasitoid *Fopius arisanus* (Sonan) and the larval parasitoid *Diachasmimorpha krawssii* (Fullaway), in larger field cages (15 × 6.4 × 2 m) reached levels close to population eradication [18].

However, this type of information has not yet been obtained for flies of the genus *Anastrepha* Schiner native to the Neotropical region. *Anastrepha ludens* stands out as one of the most economically important pests in the Americas, with a distribution range from the southern United States to Central America, attacking a wide range of marketable host fruits [19]. The control of this pest species has been carried out successfully with the application of an integrated management program [20], where augmentative releases of the exotic parasitoid *Diachasmimorpha longicaudata* (Ashmead) have been included in different regions of Mexico. The mass rearing of *A. ludens* with the purpose of applying the SIT has experienced important advances in recent years [21], since a genetic sexing strain has been developed to produce and release only sterile males in the field [22,23]. Mass rearing of parasitoids has also shown notable developments, such as the 60 million parasitized pupae/week by *D. longicaudata* in the Moscafrut facility located in Chiapas, Mexico [24], as well the fully developed methodology for the mass rearing of the pupal parasitoid *Coptera haywardi* (Oglobin) (Hymenoptera: Diapriidae) [25,26]. According to different authors [27,28,29,30], *C. haywardi* possesses several attributes that make it an effective natural enemy against fruit flies and an efficient forager, such as high host specificity and a notable ability to discriminate pupae previously parasitized by both conspecific and heterospecific parasitoid species, and thus this species has great potential as a complementary control agent against *Anastrepha* pest populations [26,30].

In the present study, we evaluated the effect of the joint application of sterile males of *A. ludens* of the genetic sexing strain Tap-7 and two parasitoid species at the field cage level. The parasitoids *D. longicaudata* and *C. haywardi* were used separately to determine their effect on the suppression of the fly populations. Based on previous work, we expected that the concurrent application of both techniques would allow a significant and differential suppression of the *A. ludens* populations under evaluation.

## 2. Materials and Methods

### 2.1. Study Areas

The joint release of sterile males of *A. ludens* (Tap-7 strain) and the parasitoid *D. longicaudata* for suppressing the fruit fly population was carried out in a chicozapote (*Manilkara zapota* L.) orchard located in Metapa, Chiapas, Mexico at 106 masl (14.83° N, 92.19′ W). The mean temperature and relative humidity recorded were 26.68 ± 0.15 °C (Min: 20.50 °C–Max: 39.60 °C) and 79.9 ± 0.44% (Min: 44.50%–Max: 94.0%), respectively (ESIS^®^Datta logger MicroLogProII Fourtec, Sydney, Australia).

Given that *C. haywardi* is a parasitoid better adapted to areas with higher altitudes, the integration of sterile males of *A. ludens* (Tap-7 strain) with the release of this parasitoid species was carried out in a mixed coffee var. Robusta (*Coffea canephora* L.) and Creole mango (*Mangifera indica* L.) orchard located at 607 masl (15°00′30.80″ N, 42°09′21.83″ W) in the Rosario Ixtal ejido, Cacahoatan, Chiapas, Mexico. The mean temperature recorded in this site was 24.58 ± 1.90 °C (Min: 15.49 °C–Max: 38.0 °C) and the mean relative humidity was 71.75 ± 7.58% (Min: 35.24%–Max: 99.96%) (ESIS^®^Datta logger MicroLogProII Fourtec, Sydney, Australia). Both areas are in the Soconusco Region of Chiapas, Mexico.

### 2.2. Biological Material

Third instar (9-day-old) larvae of *A. ludens* (standard strain) were used as the founders of the fly population within a 2 × 3 m (height × diameter) field cage. The females that emerged from this population copulated with sterile males of *A. ludens* (Tap-7 strain) that were irradiated at 80 Gy [23]. Both larvae and sterile males were obtained from the corresponding mass rearing colonies of the Moscafrut facility Plant (SADER-IICA), located in Metapa, Chiapas, Mexico.

Adults of the parasitoids *D. longicaudata* and *C. haywardi* were obtained from the rearing colonies of the Moscafrut facility and from the Biological Control Laboratory of the Technological Development Unit of the Moscafrut Program SADER-IICA. Both parasitoid species currently coexist in Mexican fields attacking *Anastrepha* spp. populations [26,30].

### 2.3. Founder Population

The evaluations were carried out in cylindrical field cages of 2 × 3 m (height × diameter) with mosquito netting with a mesh size of 0.1 mm and a lateral opening. The cages were placed under the shade of trees and the structure was supported with plastic ropes tied to the branches of the trees and protected from the rain with a 4 m^2^ nylon tent.

The founding fly population was established with 500 *A. ludens* larvae (standard strain) placed on platforms covered with a 5 cm layer of soil (30–40% humidity) and leaf litter inside the cage. Different platforms to contain the soil were used according to the species of parasitoid used. In the evaluations with *D. longicaudata*, the larvae were placed in a 0.63 m^2^ tray with soil supported by a metallic structure. Given that *C. haywardi* is a pupal parasitoid, a platform with a larger area (1.60 m^2^) was used to put the soil and the 500 larvae to allow the females to forage and oviposit on the formed pupae. Both structures were 25 cm from the ground and the supporting legs were protected with Stikem^®^ (Seabright Laboratories, Emergville, CA, USA), to prevent the attack of ants or other types of predators. Small mango trees (1.5 m high) were placed in each cage to enrich the environment of the insects.

### 2.4. Adult Fly Emergence and Survival

The emergence and survival of adult flies inside each field cage were calculated using two sentinel cages made of a 30 × 30 × 30 cm wooden structure with the lateral sides covered with mesh, while the back side had a rubber covering with a circular opening (10.5 cm in diameter), which was covered with a cylindrical container to allow manipulation inside the cage. A Petri dish base (10 cm in diameter × 0.8 cm in height) containing 100 9-day-old third instar larvae of *A. ludens* was introduced into both cages. Almost 100% of the larvae were transformed to pupa 24 h later. The pupae were kept with soil and leaf litter inside the sampling cages under the environmental conditions mentioned above. After 15 days and after adult emergence, one cage was transferred to the laboratory to determine the emergence percentage, which was used to calculate the number of sterile males to be released into the cage as in Flores et al. [31,32] to establish a sterile:fertile ratio of 10:1. The second sampling cage was kept inside the first cage with water in 100 mL glass bottles with a piece of filter paper soaked in the water and a mix of food (hydrolyzed protein: sugar, 1:3 ratio) in small mesh bags (10 × 5 cm) hung from the ceiling of the cage. This cage was used to count the number of dead flies per day, up to day 22 of emergence, to obtain the mean daily survival.

### 2.5. Treatments

*Assay with Diachasmimorpha longicaudata.* Four treatments were carried out: (1) SIT + ABC combination, (2) release of only sterile insects (SIT), (3) release of only parasitoids, and (4) control treatment without release of either sterile males or parasitoids. Each treatment was evaluated separately in a field cage. Five repetitions of each treatment were performed. In all cases, the evaluation began by placing 500 9-day-old *A. ludens* larvae (standard strain) in 25 guava fruits (*Psidium guajava* L.), which were previously cut at the apex and the pulp was removed with a spoon and mixed with 20 9-day-old *A. ludens* larvae and then introduce back into the fruit. The fruits with larvae were sealed with a thin band of parafilm paper and hung randomly from the ceiling of the cage using a 10 cm piece of galvanized wire (No. 20) with one end shaped into a hook. The piece of wire went through the distal part of the fruit and was held with a small piece of foam rubber. The fruits with larvae were exposed to 25 previously copulated 5-day-old females of *D. longicaudata* for 24 h. After this time, the larvae were extracted from the fruits with entomological forceps, washed with water, and placed on the floor of the platform inside the cages to allow them to continue their development.

*Assay with Coptera haywardi.* The treatments with the pupal parasitoid were the same as those described above. The only difference was that the 500 9-day-old *A. ludens* larvae founding the fly population were distributed directly on the ground to allow them to bury into the substrate and start the pupation process. When the pupae were 3–4 days old, 100 previously copulated 5-day-old female parasitoids were introduced, which were kept inside the cage until completing 48 h of pupal exposure.

### 2.6. Sterile Insect Releases

When the adult flies that emerged from the soil reached sexual maturity (7–8 days of age), the number of sterile males of *A. ludens* Tap-7 strain necessary to establish a sterile: fertile ratio of 10:1 was released (e.g., [32]), using as a reference the emergence percentage obtained in the sampling cage described above. When the female flies that had already mated reached 12 days of age, we introduced oviposition units consisting of green spheres (4 cm in diameter) of fucellerone (TIC GUMS^®^, White Marsh, MD, USA) that resembled fruits. These spheres were made with a mixture of 30 g of fucellerone dissolved in 1 L of boiling water and 2 mL of green vegetable dye (McKormic^®^, Mexico City, México) that was poured into molds to obtain the desired shape. The spheres were individually covered with parafilm paper and exposed to the ovipositing flies in groups of five wrapped in tulle mesh. Four of these oviposition unit sets were distributed inside each cage (total of 20 spheres), where they were kept for 24 h and then were replaced with new spheres for 10 consecutive days.

### 2.7. Parameter Determination

*Parasitism percentage.* This parameter was calculated using a sample of 100 larvae (in the case of *D. longicaudata*) or 100 pupae (in the case of *C. haywardi*) that were collected in both trials after exposure to the female parasitoids. The larvae/pupae were kept under laboratory conditions in 250 mL plastic containers with moist vermiculite and a ventilated lid (8 cm in diameter × 4.5 cm in height) until adult emergence, which occurred after 15 days in the case of *D. longicaudata* and after 30 days in *C. haywardi*. Parasitism percentage was calculated with the formula: parasitism percentage = (No. adult parasitoids) × 100/(sum of flies and adult parasitoids) [10].

*Induction of sterility.* The spheres collected from the cages were transferred to the laboratory where the fly eggs were carefully removed with a scalpel and incubated for one day in a 1 L container with water (700 mL, 8 cm in diameter × 20 cm in height), which was oxygenated by bubbling generated by a fish tank pump at 5500 Pa. Subsequently, three samples of 100 eggs were taken from each treatment per day, which were aligned with a fine brush on a piece of black nylon cloth placed inside a humid chamber (Petri dish) with a water-saturated sponge. A stereomicroscope (ZEISS^®^ Discovery V8, Oberkochen, Germany) with 5× magnification was used to count the total number of hatched eggs (empty chorions) from the aligned eggs preserved in the humid chamber until the seventh day.

### 2.8. Statistical Analysis

The data from each trial were analyzed separately. Mean parasitism percentage was compared between the ABC and ABC + SIT treatments by means of a simple ANOVA. Mean fertility was compared between the four treatments with an ANOVA as well, but a Welch Test was used due to homoscedasticity issues. Fly survival was compared between the different treatments by means of a log-rank test. An ANOVA was applied for the fruit fly emergence data in sentinel cages and for the emergence of males. For the sterile male released in the field cage, the averages between SIT and ABC + SIT were compared using a *t* test. Net fertility was obtained by multiplying survival (lx = No. of live flies/No. of flies of the original cohort) by fertility (% of hatched eggs). This value was obtained during the fly fertility evaluation period (between 12 and 22 days of age). Net reproduction per day was compared between treatments using a repeated measures ANOVA.

We used the data of survival, fecundity, and egg hatching percentage to construct a partial life table (0–51 days), which was used to calculate the population reproductive parameters of *A. ludens* in each treatment: gross fertility rate, gross hatching rate, number of eggs per female per day, mean age of net fertility, and intrinsic rate of increase. The calculations were carried out using the formulas published by Carey [33].

## 3. Results

### 3.1. Adult Fly Emergence

Adult fly emergence percentage in the sentinel cages was similar in all treatments of each trial and in all environmental conditions (*D. longicaudata*: F = 0.76, df = 19, *p* = 0.52; *C. haywardi*: F = 1.31, df = 19, *p* = 0.30) (Table 1). The parasitism percentages achieved by each parasitoid species inside the field cage caused lower emergence of males, mainly in treatments where *D. longicaudata* was released, while where only sterile flies were used, male emergencewas no different from the control treatment (F = 19.66, df = 19, *p* < 0.0001). The results with *C. haywardi* were similar (F = 8.26, df = 19, *p* = 0.001). Total male emergence was used as a reference factor for the number of male flies that were released in the ABC + SIT and ABC treatments (*D. longicaudata*, t = 1.95, df = 8, *p* = 0.08; *C. haywardi*, t = −1.80, df = 8, *p* = 0.10) (Table 1).

### 3.2. Parasitism

The parasitism percentages by *D. longicaudata* obtained in the ABC + SIT and ABC treatments were 63.36 ± 5.94 and 73.18 ± 3.71, respectively, with no statistical difference between treatments (F = 0.47, df = 9, *p* = 0.51). The parasitism percentages by *C. haywardi* were lower, with mean values of 34.98 ± 2.92 and 36.09 ± 3.18 for the respective treatments of ABC + SIT and ABC and without significant differences between treatments (F = 0.18, df = 9, *p* = 0.68; Figure 1a,b).

### 3.3. Induction of Sterility

*Assay with D. longicaudata.* The egg hatching percentages were different between treatments. The highest hatching percentage was observed in the control treatment, but a gradual reduction was observed in the treatments with only parasitoids and only sterile males. The highest induction of sterility (lowest egg hatching percentage) occurred with the joint release of parasitoids and sterile males (F = 699.10, df = 3, *p* < 0.001; Figure 2a).

*Assay with C. haywardi.* The results of this trial were similar to those of the trial with *D. longicaudata* (F = 259.18, df = 3, *p* < 0.001), since the highest egg hatching percentage was observed in the control treatment, with subsequent decreases in fertility with the exclusive release of *C. haywardi* or sterile males (Figure 2b).

### 3.4. Daily Net Reproduction

Figure 3 shows the fly fertility output of the surviving females in the 10 days of evaluation. Mean net fertility per day in the treatments related with *D. longicaudata* was very similar to total fertility and was statistically different between treatments (F = 148.86, df = 3, *p* < 0.0001). There were no differences between days (F = 2.10, df = 9, *p* = 0.15), although both conditions did interact (F = 0.02, df = 27, *p* = 0.01). The results with *C. haywardi* were similar to those obtained with *D. longicaudata*. The difference between treatments was statistically significant (F = 69.09, df = 3, *p* < 0.0001); however, there was no difference between days (F = 1.50, df = 9, *p* = 0.36) and no interaction between the two conditions (F = 0.03, df = 27, *p* = 0.47; Figure 3b).

### 3.5. Survival

In the trials with *D. longicaudata*, there were no statistical differences between treatments in the survival of fly emerged adults in the two sampling cages (χi = 0.42, df = 3, *p* = 0.93). However, in the case of *C. haywardi*, survival was higher in the control treatment than in the rest of the treatments (χi = 11.81, df = 3, *p* = 0.008).

#### Reproductive Parameters

Table 2 shows the gross fertility and egg hatching rates in each treatment including one of the two parasitoid species evaluated. Gross fertility rate decreased up to 15 and 6 times when sterile flies were combined with *D. longicaudata* and *C. haywardi*, respectively. The impact of the parasitism by *D. longicaudata* was important in the decrease of this parameter and had a stronger influence when used jointly with the SIT. Mean hatching percentage also decreased in a similar way in the treatments that included the two parasitoid species, where gross hatching rate decreased up to four times when ABC and SIT were combined. Similarly, mean number of eggs per female per day was lower in the treatments with ABC plus SIT. In this case, we observed a considerable decrease when *C. haywardi* was released when using both techniques, but the influence of the parasitism by *D. longicaudata* was more transcendental when combined with the SIT, reaching negative values. Mean age of net fertility was around 45–47 days in all treatments. The intrinsic rate of population increase of *A. ludens* was also reduced with the application of ABC and SIT separately, and the decrease was approximately one third when SIT was combined with *C. haywardi* and 50% when it was combined with *D. longicaudata*.

## 4. Discussions

The results obtained in this study show that the joint release of sterile flies and insect parasitoids caused a notable reduction in the potential increase of an *A. ludens* population. This indicates that the combination of these techniques promotes an additive response both in the case of *C. haywardi* and *D. longicaudata*, where a significant difference was observed with the latter species when compared with the ‘only sterile males’ treatment. The analysis of the *A. ludens* reproductive data at a population level showed significant decreases in the fertility and fecundity of females when both techniques were applied, which directly affected the intrinsic rate of increase in a synergistic manner. This showed that, if both techniques are used appropriately, they can contribute efficiently and harmoniously to a sustainable and ecologically oriented control of this type of pests.

Both species of parasitoids exerted a suppressive effect on the pest population that was reflected in the high parasitism percentages achieved, both at the larval and pupal level (Table 1, Figure 1), which indicates that the released sterile insects encountered a population with decreased fertility and with a more favorable sterile:fertile ratio. According to the population reproduction parameters of *A. ludens*, the observed decreases had greater consequences when the two techniques were applied. In the case of the treatment with the parasitoid *D. longicaudata*, the expected number of eggs per fly female becomes negative, which indicates that continuous parasitoid and sterile fly release events would result in an increasing ratio of sterile insects that would theoretically tend to eradicate the fly population. Since Knipling’s initial proposal in 1992 [34], which emphatically recommended the augmentative release of parasitoids as a suppression element, different studies at the field level have shown that it is possible to achieve notable suppression in different host-parasitoid relationships. Examples of the above are the cases of *D. tryoni* with *C. capitata* in Hawaii and Guatemala [16,35], or *D. longicaudata* in Argentina [13] and Spain [36]. The same has been found in *Anastrepha* species of economic importance with *D. longicaudata* [11,12], a parasitoid species that has become an important biological control agent in different parts of the world. Similar results have also been obtained with flies of the genus *Bactrocera* in Asia with the release of *F. arisanus* or *Psyttalia concolor* (Szépligeti) [37], which are braconid parasitoids of the Subfamily Opiinae that use larvae as hosts. In the case of *C. haywardi*, there are high expectations for its use as a biological control agent against flies of the genus *Anastrepha* [27,29], which are based on two important characteristics: its high specificity and its high capacity to discriminate larvae previously parasitized by conspecifics or heterospecifics [26]. In addition to the above, our results suggest that this species could be integrated as a complementary suppression element in IPM schemes that use SIT as a central pest control strategy.

There is enough information supporting the SIT as a highly efficient element against fruit fly pest populations in different worldwide scenarios, such as the containment, suppression, or eradication of wild populations [38]. Emblematic examples are the cases of the Preventive Release Programs (PRP) in Florida and California for *C. capitata* [39], the Medfly eradication program in the Mexico-Guatemala border [5,6] and the Dominican Republic [40], and the suppression programs in Israel, Spain, and Argentina [41], to name a few. In Mexico, sterile insects of the *A. ludens* Tap-7 strain have been operationally released in different states of the country, representing a substantive pest control strategy [9,21]. In the citrus region of the state of San Luis Potosí, the fly/trap/day (FTD) index decreased drastically with the use of SIT, and thus aerial spraying was eliminated, which implied a significant reduction in the amount of control chemicals traditionally used against this pest, as well as significant increases in the production of healthy and high-quality fruit [21]. In the state of Chiapas, Mexico important benefits have also been obtained from the use of this technique, which have been reflected in a decrease in the use of the more traditional chemical control, an increase in pollinator populations associated with mango orchards, and a sustained trend of decreased larval batches in mangoes for export [9].

Theoretical models associated with the combined use of these techniques demonstrate a synergistic effect in the suppression of the pest population [1,2]. This effect is attributed to the fact that both techniques do not overlap in their action and their particular effects contribute mutually to reducing the population dynamics of the pest. Parasitoids in this case are considered highly specific species that efficiently search for their hosts [42]. Our data showed that the fly population was significantly reduced by parasitism, with percentages ranging from 30 to 70, depending on the species of parasitoid used, which increases the probability of wild females mating with sterile males, thus achieving a higher percentage of induced sterility in the target population. This implies that, in a closed environment and in a single generation including both control strategies, around 20% (with *C. haywardi*) and 10% (with *D. longicaudata*) of the *A. ludens* population would remain to breed in the next generation, which shows that these strategies are mutually compatible and highly eligible under an integrated pest management (IPM) approach [17]. The intrinsic rate of increase showed a tendency to decrease with the application of either ABC or SIT; however, the effect on this parameter was more transcendental in the population dynamics when the combined action of the two techniques was used.

The combined application of ABC and SIT against fruit flies can represent a clear perspective for a sustainable control of these pests for governments and productive societies, especially in the current scenario of global warming, climate change, and threats to biodiversity. Undoubtedly, both technologies are conceived as highly environmentally friendly options with solid theoretical and experimental support [43]. However, both techniques are subject to a series of factors in the natural environment that may affect their expected theoretical performance. In the case of parasitoids, one factor may be hosting availability, which involves the appropriate age of the host as well as the shape and size of the fruit where it is found, since it can operate as a physical refuge for the larva [44,45]. *Anastrepha ludens* commonly oviposits in large fruits such as citrus or mango, where the larvae can protect themselves from parasitoid attacks and reduce their chances of being parasitized [20,46]. Similarly, sterile males face different challenges when they are deployed into the field. The most common one is successfully competing with wild males for mattings with females available in the environment, which in turn becomes a significant challenge for managers of fruit fly mass-rearing facilities. A more limited dispersal and survival capacity of sterile insects has also been observed with respect to wild populations. One way to compensate for these disadvantages is to release sterile insects at densities high enough to achieve reasonable sterility in the wild population [32,33,47,48]. Another associated factor limiting the suppressive effect of both techniques is the continuous invasion of flies from surrounding areas, which may vary depending on the fruit production season in the working area. In this regard, various authors have emphasized that these techniques should be applied at the regional level (e.g., [35,49]), since the suppressive effect in small areas can be minimized by the presence of flies coming from surrounding areas. One way to combat this is the formation of buffer areas, where the invading population is suppressed to reduce the flow of wild flies towards the area of commercial orchards [9].

Our results showed that the joint use of ABC and SIT on a population of *A. ludens* has a direct additive effect plus an additional effect that can be characterized as synergistic within the population dynamics of the flies with the periodic releases of both types of insects. This effect can be of paramount importance for the suppression or eradication of populations of these types of pest, coupled with the advantage of the low ecological impact that characterizes both techniques.

## Figures and Tables

**Figure 1 insects-14-00337-f001:**
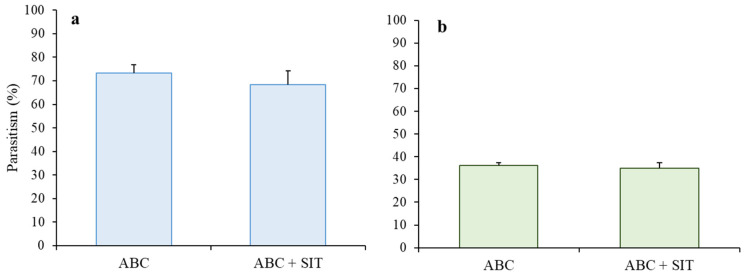
Parasitism percentages in the treatments of ABC and ABC + SIT (**a**) *D. longicaudata*, (**b**) *C. haywardi*. Lines above bars represent the standard error. There were no statistical differences between treatments in each trial (ANOVA).

**Figure 2 insects-14-00337-f002:**
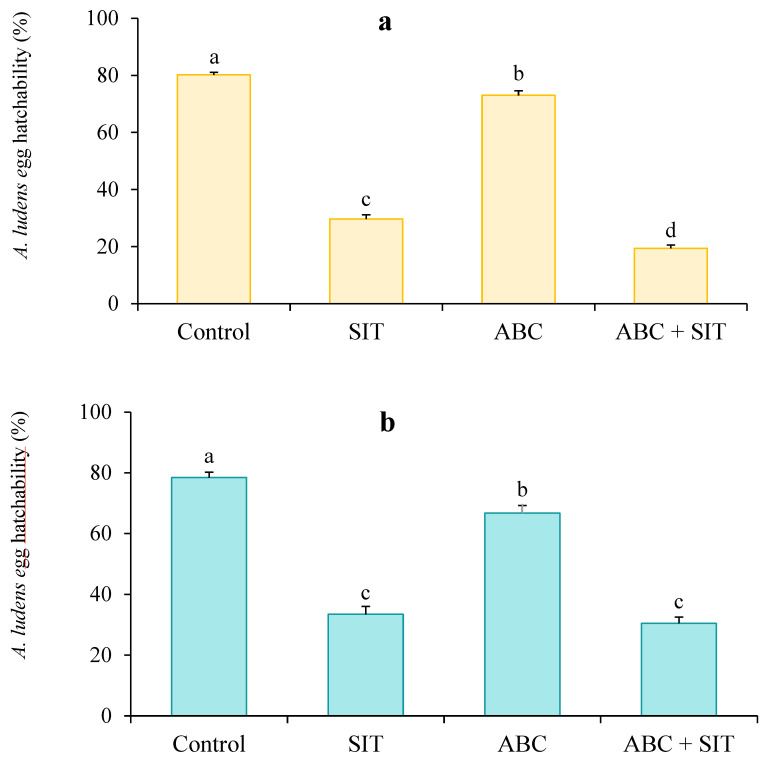
Egg hatching percentages of *A. ludens* obtained from four populations treated with parasitoids at the larval stage and sterile male flies (Tap-7). (**a**) Treatments with *D. longicaudata* as a parasitoid, (**b**) Treatments with *C. haywardi* as a parasitoid. Lines above bars represent the standard error. Different letters above bars indicate statistical differences (ANOVA, Welch Test).

**Figure 3 insects-14-00337-f003:**
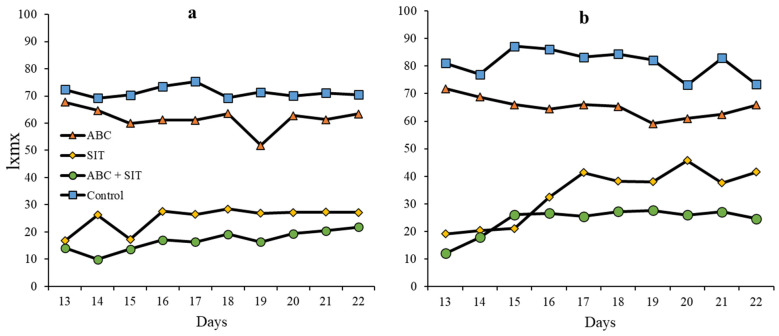
Net reproduction per day of *A. ludens* treated with parasitoids and sterile male flies, (**a**) Treatment with *D. longicaudata* as a parasitoid, (**b**) Treatment with *C. haywardi* as a parasitoid.

**Table 1 insects-14-00337-t001:** Means (±SE) of fly emergence (sentinel cages), male emergence (field cages), number of sterile males released, and the sterile:fertile ratio with ABC (*D. longicaudata* or *C. haywardi* parasitoids) plus SIT on an *A. ludens* population at field cage level.

Treatments	Emergence (%) in Sentinel Cages	Males Emerged in Field Cages	Sterile Males Released	Rate Sterile: Fertile
*Assay with D. longicaudata*				
ABC	85.40 ± 3.18 a	63.43 ± 10.61 b	--	--
SIT	80.40 ± 2.73 a	238.00 ± 3.39 a	2250.00 ± 7.90 a	10:1
ABC + SIT	80.20 ± 2.47 a	73.76 ± 16.36 b	2136.00 ± 57.73 a	29:1
Control	82.40 ± 2.73 a	239.00 ± 7.48 a	--	--
*Assay with C. haywardi*				
ABC	74.80 ± 6.39 a	117.74 ± 5.05 b	--	--
SIT	75.80 ± 4.27 a	193.00 ± 11.68 a	1896.0 ± 106.8 a	10:1
ABC + SIT	85.80 ± 3.49 a	116.31± 18.89 b	1716.0 ± 69.97 a	15:1
Control	82.20 ± 3.48 a	214.00 ± 15.60 a	--	--

Average values followed by different letter in the same column implies statistical difference. One-way ANOVA and Tukey test.

**Table 2 insects-14-00337-t002:** Reproductive parameters of a closed population of *A. ludens* subjected to parasitism by augmentative biological control (ABC) or the release of sterile flies to induce sterility (SIT) and the combination of both techniques (ABC + SIT) in field cage conditions.

Treatments	Gross Fertility Rate	Gross Hatch Rate	Eggs per Female per Day	Mean Age Net Fertility (Days)	Intrinsic Rate of Increase (r)
*Assay with D. longicaudata*					
ABC	80.59	73.24	0.04	46.47	0.094
SIT	145.46	30.59	1.06	47.07	0.105
ABC + SIT	27.34	20.37	−0.29	47.53	0.069
Control	424.14	80.29	3.64	46.56	0.129
*Assay with C. haywardi*					
ABC	144.30	71.41	1.01	45.63	0.108
SIT	117.00	39.84	1.09	47.18	0.100
ABC + SIT	63.65	26.85	0.42	46.72	0.088
Control	433.05	84.21	3.67	46.17	0.131

## Data Availability

The raw data used for this manuscript were up-loaded to Zenodo under https://doi.org/10.5281/zenodo.7709513.

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
