# Peer review of "Additive Effect of Releasing Sterile Insects Plus Biocontrol Agents against Fruit Fly Pests (Diptera: Tephritidae) under Confined Conditions"

_insects, 2023, doi:10.3390/insects14040337_

Round 1

Reviewer 1 Report

The article reports on interesting experimental data that confirm the additive value of combining control strategies targeting different stages of the pest. Results reflect what can be reasonably expected but the experimental experimentation is always important to demontrate it. The article is well written and deserves publication. I appreciate the fact that discussion takes into account the differences between confined experiments and open field allowing the reader to contextualize the results in a broader frame.

I can list just minor required revisions.

I am not convinced about the opportunity to statistically compare the experiments differing for the parasitoid species. Indeed, the parasitoid species are different but also the experimental systems and the environments are different. I suggest to only compare treatments related to each single parasitoid species. As a consequence, letters in figure 1 should be assigned within a single experiment and only comparing the treatments.

The text in tables and figures should be checked because the language requires revision (especially tables 1 and 2).

Lines. 269-270: “The lowest egg hatching percentage also occurred with the simultaneous release of C. haywardi parasitoids and sterile flies (Figure 2b).” This statement should be avoided if difference is not statistically significant.

Lines 349-350: “F. arisanus or Psyttalia concolor (Szépligeti) 349 [38], which are braconid parasitoids of the Subfamily Opiinae characterized by having large ovipositors and using larvae as hosts”. It is not really important in the context of this article, but if you want to be precise, F. arisanus has a thin ovipositor as it oviposits in host eggs not larvae. Then, development occurs in the larvae and puparia of the hosts (Moretti et al 2003).

REF: Moretti, R., e M. Calvitti (2003). Mortality by parasitization in the association between the oo-pupal parasitoid Fopius arisanus and Ceratitis capitata. BioControl, 48(3): 275-291.

Author Response

Response to Reviewer 1, Insects MS 2304187

The article reports on interesting experimental data that confirm the additive value of combining control strategies targeting different stages of the pest. Results reflect what can be reasonably expected but the experimental experimentation is always important to demontrate it. The article is well written and deserves publication. I appreciate the fact that discussion takes into account the differences between confined experiments and open field allowing the reader to contextualize the results in a broader frame.

R: We highly appreciate the comments and opinion of reviewer 1.

I can list just minor required revisions.

I am not convinced about the opportunity to statistically compare the experiments differing for the parasitoid species. Indeed, the parasitoid species are different but also the experimental systems and the environments are different. I suggest to only compare treatments related to each single parasitoid species. As a consequence, letters in figure 1 should be assigned within a single experiment and only comparing the treatments.

R: Actually, we analyzed the data for each treatment by each parasitoid species separately (see lines121-122 in the statistical section). However, made some changes in the figure 1 in the position and nomenclature of used letters to avoid confusions. Also, we add some clarification in the pie of figure 1.

The text in tables and figures should be checked because the language requires revision (especially tables 1 and 2).

R: Reviewer is right and we apologize, it seems that we include a preliminary version of the tables’ titles. We corrected both titles.

Lines. 269-270: “The lowest egg hatching percentage also occurred with the simultaneous release of C. haywardi parasitoids and sterile flies (Figure 2b).” This statement should be avoided if difference is not statistically significant.

R: we agree and delete this part of the text.

Lines 349-350: “F. arisanus or Psyttalia concolor (Szépligeti) 349 [38], which are braconid parasitoids of the Subfamily Opiinae characterized by having large ovipositors and using larvae as hosts”. It is not really important in the context of this article,

R: reviewer is right, the comment on the of the size of the ovipositor is not relevant in the context of this article. We delete this part.

but if you want to be precise, F. arisanus has a thin ovipositor as it oviposits in host eggs not larvae. Then, development occurs in the larvae and puparia of the hosts (Moretti et al 2003).REF: Moretti, R., e M. Calvitti (2003). Mortality by parasitization in the association between the pupal parasitoid Fopius arisanus and Ceratitis capitata. BioControl, 48(3): 275-291.

Reviewer 2 Report

Dear Authors,

This manuscript provides substantially enough evidence regarding the synergistic/sustainable action of insect parasitism and SIT in a major fruit fly species.
Bibliography, methodology and results are very well documented and supported using acceptable/very good use of English language.
The effect of each biological factor was wisely examined seperately, but also in combined effectiveness to achieve their possible effect levels..
Satistical analysis and discussion are also given thorouhly based on major bibliographic references (see only one comment in the text concerning statistics).
Given that only a few minor grammatic/syntax/formatting comments should be addressed in the pdf, then, in my opiniion, this ms could be published in the current form.

Author Response

Insects MS 2304187, response to Reviewer 2

Dear Authors,
This manuscript provides substantially enough evidence regarding the synergistic/sustainable action of insect parasitism and SIT in a major fruit fly species.
Bibliography, methodology and results are very well documented and supported using acceptable/very good use of English language.
The effect of each biological factor was wisely examined seperately, but also in combined effectiveness to achieve their possible effect levels..
Satistical analysis and discussion are also given thorouhly based on major bibliographic references (see only one comment in the text concerning statistics).

Given that only a few minor grammatic/syntax/formatting comments should be addressed in the pdf, then, in my opiniion, this ms could be published in the current form.

R: We highly appreciate the comments and opinion of reviewer 2.

Comments on the pdf revised.

Line 13 Meaning? eg geographically isolated? ot her) please clarify

R: We used the term closed population to specify that both trials were done under field cage conditions. We change closed by confined. See line 13.

Line 128. Please state whether or not any samplings in the field had been performed for the existence of natural populations of the 2 parasitoids prior to lab reared releases. This might have lead to a different interpretation on the parasitism results.

R: we add some information to better support the use of these species. Please see lines 130 – 132-

Line 217, Unclear, please clarify or rephrase.

R: we add more information in order to be clearer. See line 219.

Line 218 Statistical Analysis, please state somewhere in this paragraph whether was any need for data transformation prior to analyses.

R: It was not necessary to use data transformation, we had homoscedasticity problems but we decided to resort to Welch's tests because there were not major statistical problems. Fortunately, all the data had normality and the other Anova conditions were satisfactorily covered.

Lines 235, 236, as suggested, we change to Adult fly in both lines

R: Done

Line 241

R: done, we change to italic the name of D. longicaudata

Figure 2, according to the suggestion of the reviewer, we change the legend of the y axes.

Table 2, we corrected the title of this table.